# Mechanism of the Affinity-Enhancing Effect of Isatin on Human Ferrochelatase and Adrenodoxin Reductase Complex Formation: Implication for Protein Interactome Regulation

**DOI:** 10.3390/ijms21207605

**Published:** 2020-10-14

**Authors:** Pavel V. Ershov, Alexander V. Veselovsky, Yuri V. Mezentsev, Evgeniy O. Yablokov, Leonid A. Kaluzhskiy, Anastasiya M. Tumilovich, Anton A. Kavaleuski, Andrei A. Gilep, Taisiya V. Moskovkina, Alexei E. Medvedev, Alexis S. Ivanov

**Affiliations:** 1Institute of Biomedical Chemistry, 10 Building 8, Pogodinskaya Street, 140006 Moscow, Russia; veselov@ibmh.msk.su (A.V.V.); yuri.mezentsev@ibmc.msk.ru (Y.V.M.); evgeniy.yablokov@ibmc.msk.ru (E.O.Y.); leonid.kaluzhskiy@ibmc.msk.ru (L.A.K.); alexei.medvedev@ibmc.msk.ru (A.E.M.); alexei.ivanov@ibmc.msk.ru (A.S.I.); 2Institute of Bioorganic Chemistry NASB, 5 Building 2, V.F. Kuprevich Street, 220141 Minsk, Belarus; tumilovicham@iboch.by (A.M.T.); bio.kovalevs@iboch.by (A.A.K.); agilep@iboch.by (A.A.G.); 3Far East Federal University, FEFU Campus, 10 Ajax Bay, Russky Island, 690922 Vladivostok, Russia; moskovkina.tv@dvfu.ru

**Keywords:** ferrochelatase, adrenodoxin reductase, isatin, in silico, affinity, complex formation, heterodimerization, surface plasmon resonance

## Abstract

Isatin (indole-2, 3-dione) is a non-peptide endogenous bioregulator exhibiting a wide spectrum of biological activity, realized in the cell via interactions with numerous isatin-binding proteins, their complexes, and (sub) interactomes. There is increasing evidence that isatin may be involved in the regulation of complex formations by modulating the affinity of the interacting protein partners. Recently, using Surface Plasmon Resonance (SPR) analysis, we have found that isatin in a concentration dependent manner increased interaction between two human mitochondrial proteins, ferrochelatase (FECH), and adrenodoxine reductase (ADR). In this study, we have investigated the affinity-enhancing effect of isatin on the FECH/ADR interaction. The SPR analysis has shown that FECH forms not only homodimers, but also FECH/ADR heterodimers. The affinity-enhancing effect of isatin on the FECH/ADR interaction was highly specific and was not reproduced by structural analogues of isatin. Bioinformatic analysis performed using three dimensional (3D) models of the interacting proteins and in silico molecular docking revealed the most probable mechanism involving FECH/isatin/ADR ternary complex formation. In this complex, isatin is targeted to the interface of interacting FECH and ADR monomers, forming hydrogen bonds with both FECH and ADR. This is a new regulatory mechanism by which isatin can modulate protein–protein interactions (PPI).

## 1. Introduction

Isatin (indole-2,3-dione) is an endogenous oxidized indole exhibiting a certain spectrum of biological and pharmacological activities, including regulation of gene expression, neuroprotection, antitumor action, etc. (for review see [1]). This compound has been found in various tissues and biological fluids of mammals, where its concentration varies from 0.1 μM to 10 μM, and in the case of its administration in vivo, isatin concentration in certain organs (e.g., the brain) can reach 50–100 μM [1,2]. These numerous biological and/or pharmacological effects are obviously mediated by isatin interaction with many isatin-binding proteins, identified during proteomic profiling of brain preparations of mice and rats [3,4,5]. However, besides direct action of isatin on its particular targets, characterized by classical binding constants and altered biological functioning of individual proteins [6,7] it can also act as a bidirectional regulator of protein–protein interactions (PPI) increasing or decreasing affinity of interacting protein partners [1,8]. This novel regulatory role of isatin has been originally demonstrated by us for the interaction of two pairs of proteins: (1) ferrochelatase (FECH) and adrenodoxin reductase (ADR) [8], (2) thromboxane synthase (TBXAS1) and cytochrome P450 2E1 (CYP2E1) [9]. It was particularly interesting that individual proteins of each investigated pair of the interacting proteins demonstrated low (if any) isatin binding capacity, but the addition of isatin caused a several-fold increase in the affinity of their interaction [8,9].

Our previous SPR study showed that a pharmacologically relevant concertation of isatin (100 µM) significantly increased affinity of the FECH/ADR complex: the K_d_ value decreased from 13 µM to 2 µM [8]. These isatin-induced changes in ∆K_d_ included both an increase in the FECH/ADR association rate (the k_on_ value demonstrated a four-fold increase) and a less than 2-fold decrease in the dissociation rate constant (k_off_) [8]. At a higher pharmacological concentration of isatin (270 μM) this effect became even more pronounced (the change in the K_d_ value was more than 15 times). We also found that isatin did not interact with each individual protein (FECH, ADR) immobilized on the optical chip. However, isatin had no effect on other interactions of FECH with cytochrome b5 and SMAD4 thus indicating specificity of its effect on the FECH/ADR interaction [8].

In this study, using a surface plasmon resonance (SPR) based biosensor we have investigated the specificity and molecular mechanism responsible for the affinity-enhancing effect of isatin on the interaction of FECH and ADR. Since FECH can exist in monomeric and dimeric forms [10,11], we have initially investigated which complexes (dimeric or trimeric) FECH formed during interaction with ADR, and other tested proteins (cytochrome *b*_5_ type B (CYB5B) and SMAD family member 4 (SMAD4)) [12,13,14]. Experiments have shown that CYB5B and SMAD4 can interact with both the monomeric and dimeric forms of FECH, with the formation of heterodimeric and heterotrimeric complexes, respectively. At the same time, ADR interacted only with the monomeric form of FECH (with formation of heterodimers). Individual proteins, FECH and ADR, immobilized on the optical chip, did not interact with isatin and its several derivatives. The effect of isatin on the FECH/ADR complex formation was specific, as several isatin analogues tested had no influence on the complex formation. Using computer-aided molecular modeling, the hypotheses of the most probable structures of the FECH/isatin/ADR ternary complexes were constructed. Their comparison with experimental data made it possible to choose the most probable model of such ternary complex. This model reasonably explained the affinity-enhancing effect of isatin on the pair of interacting proteins tested.

## 2. Results

### 2.1. Interaction of FECH with ADR

Since FECH can exist both in monomeric and in dimeric forms [10,11], we have performed a comparative SPR analysis of the interaction of ADR with FECH monomers and dimers, immobilized on the optical chip. In accordance with our previous study [8] immobilized FECH monomers interacted in the concentration-dependent manner with added ADR (Figure 1A). In another experiment FECH solutions were injected to verify the ability of immobilized FECH monomers to participate in homodimerization (Figure 2). We also carried out sequential injections of FECH and ADR protein solutions through all four biosensor channels (“empty channel” Fc1 (control 1); channel Fc2 with FECH monomers; channel Fc3 with FECH monomers treated with NHS/EDC mixture (control 2); channel Fc4 with FECH dimers treated with NHS/EDC mixture).

Experiments have shown that ADR interacted almost exclusively with the FECH monomers compared to dimers (Figure 1B). 

It is especially important that ADR effectively interacted with its functionally significant protein partner adrenodoxin (Adx) (Appendix A). This suggests functional binding competence of ADR. The low level of ADR binding in the Fc4 channel with the FECH dimers could be explained by contaminations with FECH monomers in the channel. The presence of small amounts of monomers after the chemical stabilization of the dimeric form of other proteins was also noted earlier [15,16].

### 2.2. The Effect of Isatin and Its Derivatives on FECH Interaction with Protein Partners

In order to assess specificity of the isatin effect on PPI, we have investigated the effect of seven derivatives of isatin (Table 1), which differ in the key positions of the isatin molecule for the interaction with these proteins.

In order to confirm our previous data [8], we have initially investigated the absence of significant interaction of these compounds individually with the monomeric form FECH and ADR immobilized on a chip. In these experiments two other proteins, B2M and BSA, were also used (Figure 3). 

Figure 3 shows, that no noticeable binding of isatin (at the concentration achievable in vivo after administration of a pharmacologically relevant dose [1,2,5]) and its derivatives (except for 5-bromoisatin and 5-iodoisatin) was observed. For an additional “inverted” control, 5-aminoisatin was immobilized in one of the biosensor channels and samples of eight different recombinant proteins, including FECH and ADR, were injected. The binding signals of FECH and ADR to immobilized 5-aminoisatin slightly exceeded the binding range of the other six control proteins to 5-aminoisatin (Appendix A and see also [17]). Next, a comparative SPR analysis of the possible effect of isatin derivatives on FECH homodimerization as well as FECH/ADR heterodimerization was performed, using the protocol described in our earlier work [8]. We did not find significant differences in the kinetic and affinity parameters of FECH/ADR and FECH/FECH complex formation in the presence of seven different isatin derivatives as compared to control (without isatin derivatives) (Table 2), whereas isatin induced complexation of FECH/ADR (Appendix A and also [8]), but did not affect the formation of FECH/FECH dimers (Table 2). Although parameters of complex formation between FECH (cross-linked dimer) and ADR (Figure 1B) were not determined, but SPR protein binding signals in the presence of isatin remained unchanged. Thus, based on the data obtained, it can be assumed that the binding site of isatin in the FECH/ADR heterodimer is at the interface between two proteins and is not accessible for the binding of isatin derivatives used in this work. This result indicates the specificity of the action of isatin as a PPI regulator.

### 2.3. Molecular Modeling of the Isatin Effect on the Affinity of Protein–protein Interaction

It was logical to assume that the affinity-enhancing effect of isatin on the pair of interacting proteins lacking individual isatin-binding sites (FECH and ADR) would involve the FECH/ADR heterodimer interface. In order to assess possibility of such scenario we performed computer modeling of possible configurations of molecular complexes formed. Figure 4 shows the algorithm of modelling. ClusPro, GRAMM-X, and ZDock. Among these programs ClusPro and ZDock present the values of scoring functions, while GRAMM-X presents a list of top 10 most probable complexes. Based on the scoring function of the ZDock output file (see Appendix A), 10 top ZDock possible complexes were selected. Since the ClusPro program calculates several scoring functions, two sets of 10 models with the best “balanced” and “hydrophobic” scoring functions were selected (Table 3). Thus, 40 models of possible complexes (10 from GRAMM-X, 10 from ZDock, and 20 from ClusPro) were selected for the subsequent computer-aided analysis.

As shown above, the contact surface of FECH monomers is involved not only in the formation of homodimers (FECH/FECH) but also in formation of FECH/ADR heterodimers. Analysis of molecular models obtained using molecular docking showed that in 24 of 40 models of FECH/ADR complexes (ClusPro_balance-8, ClusPro_hydrophobic-3, GRAMM-X-3, ZDock-10) the interface of FECH in these complexes was overlapped with the interface in the FECH/FECH complex. For further selection of the most probable protein complexes, the contact areas of interacting proteins and the number of hydrogen bonds that could be formed between the interacting proteins were calculated. The contact area in the complexes varied from 673 Å2 to 1412 Å2, and the number of hydrogen bonds from 1 to 25. Based on these data, 8 from 24 models of complexes were selected. These included ClusPro_balance-3, ClusPro_hydrophobic-1, GRAMM-X-1, ZDock-3 with contact areas ranged from 1056 Å2 to 1412 Å2 and the number of hydrogen bonds from 9 to 25.

In order to find the most probable binding site involved in the isatin interaction with the FECH/ADR complex, molecular docking of isatin and its seven derivatives was performed over the entire surface of each of the 8 models of the complex. Data interpretation (and selection of the most appropriate model) was based on the SPR data (Figure 3) suggesting the lack of direct isatin interaction with the putative binding site.

Analysis of the results has shown that only one model satisfying all these criteria is possible: it includes the formation of the FECH/isatin/ADR ternary complex (Figure 5A). This model is based on one of the models obtained as a result of using the GRAMM-X program. In this ternary complex, isatin is located at the FECH contact interface with ADR and it interacts with both proteins with formation of two additional hydrogen bonds (one with each interacting protein). Other isatin derivatives do not bind at this site. It seems that additional substituents in compounds 2-7 have prevented their accommodation in the small cavity at the complex interface. Isatin interacts with the FECH residues THR229, GLN278, ALA282, GLN285 and LYS286 (and forms the H-bond) and also with ADR the residues ARG84 (forms the H-bond through oxygen of the main chain), ASP85 and ARG417 (Figure 5B). In the cell FECH/ADR heterodimers are obviously temporary complexes, which are formed during dissociation of the FECH dimer. Nevertheless, such complex formation may involve 13 hydrogen bonds and 10 salt bridges. The binding of increased concentrations of endogenous or exogenously administered isatin will stabilize these temporary complexes.

## 3. Discussion

Using human FECH and ADR complex formation as a model system, in this study, we have considered a fundamentally new mechanism of the affinity-enhancing effect of isatin.

This mechanism involves formation of a ternary complex protein/isatin/protein in which each of the interacting proteins does not bind isatin or binds it weakly. Isatin influenced only heterodimer formation by interacting with FECH and ADR monomers. At physiological concentrations isatin has no effect on such PPI. However, after administration of a pharmacologically relevant dose its intracellular level may significantly increase (such scenario has been considered in [1]). This suggests that at pharmacologically relevant concentrations this regulator will increase the affinity of some PPIs and thus change the interactome. Such mechanism provides reasonable explanation of the altered profile of isatin binding proteins observed after administration of a high neuroprotective dose of isatin to animals [18]: appearance of isatin-binding proteins specific for isatin-treated animals due to the formation of new clusters of PPI and/or novel ligand-induced binding sites.

According to the crystal structure, the adrenodoxin (ADX) complex formation with ADR, involves Asp79, Asp76, Asp72, and Asp39 of ADX and Arg211, Arg240, Arg244, and Lys27 residues of ADR, respectively [19]. The molecular binding site predicted in our studies by molecular modeling for isatin interaction with ADR is located outside the ADR binding region involved in the interaction with ADX, a canonical redox partner of mitochondrial cytochrome P450 (CYP).

It is known that FECH is involved in formation of various protein complexes with other proteins, for example, with frataxin [20] or with progesterone receptor membrane component 1 (PGRMC1) and PGRMC2 [21]. However, these interactions involve FECH surface sites other than the contact region responsible for interaction of FECH monomer with each other. In our experiments, in the absence of isatin, the low affinity FECH/ADR complex (K_d_ of about 15 μM) did not compete with the classical FECH/FECH dimerization (K_d_ of about 0.5 μM). However, in the presence of increasing isatin concentrations the equilibrium shifted towards formation of a higher affinity FECH/ADR complex (K_d_ ≤ 1 μM) [8]. In this case, isatin can act as a molecular switch for the physiological reprogramming of the protein complex formation.

Although the functional importance of FECH/ADR heterodimerization remains unknown, certain evidence exists for indirect functional associations between FECH and ADR. For example, like FECH, the protein pair ADR/ADX is also involved in the final stages of heme synthesis from protoporphyrin [22,23,24,25]. ADR acts as an electron carrier from NADPH to its redox partner, mitochondrial ferredoxin (adrenodoxin, ADX), thus participating in the reduction of 2Fe-2S clusters [26]. In this context, it is possible that the presence of the 2Fe-2S cluster in FECH [11] determines interaction of this enzyme with ADR. Thus, the same intracellular localization of FECH and ADR [27,28], as well as ADR association with the functioning of enzymes containing 2Fe-2S clusters (FECH) may indicate that FECH and ADR are involved in the same biochemical pathway—the final stages of heme biosynthesis from protoporphyrin. These facts support the presence of a functional relationship between FECH and ADR, which may explain the fact of the intermolecular FECH/ADR interaction, which we found using an SPR biosensor. The biological role of the affinity-enhancing effect of isatin on FECH/ADR complex formation thus needs further investigation.

## 4. Materials and Methods 

### 4.1. Recombinant Proteins and Chemicals

Highly purified preparations (>95% by SDS-PAGE) of human NADPH-dependent adrenodoxin reductase (ADR), mitochondrial cytochromes *b_5_* (CYB5B) were expressed and purified in the Institute of Bioorganic Chemistry NASB (Minsk, Belarus). Methods of their expression in *E. coli* cells as His-tagged proteins, their isolation, purification and functional analysis have been described earlier [29,30,31]. SMAD4 and FECH were expressed in *E. coli* as His-tagged proteins and purified to homogeneity by means of metal-affinity chromatography followed by chromatography on hydroxyapatite and dialysis [8]. Preparation of 5-aminoisatin was synthesized as described in [32]. The micrOTOF-Q II (Bruker, Bremen, Germany) mass-spectrometer was used for quality control of protein preparation according standard protocol of protein identification. Isatin derivatives were synthesized using standard chemical procedures in Far East Federal University (Vladivostok, Russia). Beta-2-microglobulin (B2M) and bovine serum protein (BSA) were purchased from USBio (Cat. No. M3890-17) (Salem, MA, USA) and Calbiochem (Lot 126609) (San Diego, CA, USA), respectively.

### 4.2. Surface Plasmon Resonance (SPR) Analysis

SPR analyses were carried out at 25 °C using the optical biosensors Biacore T-200 and research grade sensor chips CM5 (GE Healthcare, Chicago, IL, USA). Optical biosensor Biacore 8K (GE Healthcare, Chicago, IL, USA) was also used for comparative assessment of binding of isatin and isatin derivatives with immobilized proteins on the optical chip surface. Running buffer HBS-EP+ (150 mM NaCl, 3 mM EDTA, 0.05% Tween-20, 10 mM HEPES, pH 7.5) with 2 mM dithiothreitol was used at a flow rate of 5 μL/min. The interaction of molecules was recorded as sensorgrams, representing the time-dependent change in biosensor signal in RU (resonance units, 1 RU is equal to 1 pg material bound on chip surface). The values of rate constants k_on_ and k_off_ were calculated in the separate mode using the BIAevaluation 4.1 software (GE Healthcare, Chicago, IL, USA) and the Langmuir binding model (1:1 complex formation). Complex dissociation constant (K_d_) values were calculated as the ratio: K_d_ = k_off_/k_on_. The SPR measurements were performed using the equipment of “Human Proteome” Core Facility of the Institute of Biomedical Chemistry (Moscow, Russia).

#### 4.2.1. FECH Immobilization on the SPR Chip

FECH was used as an immobilized ligand; its covalent binding to the chip surface was performed in accordance with the standard covalent amino coupling protocol described in the Sensor surface handbook (GE Healthcare, Chicago, IL, USA). All immobilization steps were carried out at a flow rate of 5 μL/min. After chip surface activation by injecting a mixture (1:1) of 50 μL EDC and NHS (25 μL 400 mM EDC and 25 μL 100 mM NHS) for 7 min the solution of FECH (50 μg/mL in 10 mM sodium acetate, pH 4.0) was then injected into the activated flow cell. Unreacted groups on the CM5 chip surface were blocked with 1 M ethanolamine-HCl (pH 8.5). FECH was immobilized in flow channels Fc2, Fc3 and Fc4 at the level 2400 ± 100 RU. The blank signal from the FECH-free channel Fc1 (without the immobilized FECH) was used for correcting non-specific binding of analytes to the dextran matrix of the chip. It was subtracted from the raw data, obtained from the flow cells Fc2, Fc3 and Fc4.

#### 4.2.2. Chemical Stabilization of FECH Dimers on the Optical Chip Surface

Since FECH exists as a homodimer [10,11], its complex formation with ADR could be attributed to appearance of both ADR/FECH heterodimers and ADR/FECH/FECH heterotrimers. In order to clarify this point, we have carried out SPR experiments with immobilized monomers and stabilized dimers of FECH.

Dissociation of FECH dimer complexes was prevented by the method of chemical stabilization, which was previously successfully used in a number of studies for stabilization of various protein dimers on the surface of SPR chips [15,16,33]. The main steps for preparation of the model system of the FECH monomer-dimer on an optical biosensor chip are shown schematically in Table 4. After the immobilization of FECH in three channels of the biosensor (Fc2, Fc3 and Fc4), the channels were washed with 5 mM NaOH for 15 s at a flow rate 25 μL/min for complete dissociation of protein complexes. This resulted in the decrease in the biosensor signal level in these channels from 2400 RU to 2200 RU. Subsequent experiments have shown that FECH immobilization with injections of various known reagents that destroy oligomeric complexes of proteins immobilized on a chip (100 mM mercaptoethanol; 4 M NaCl; 0.12% SDS, 10 mM glycine buffer, pH 1.5) the lack of the further decrease in the level of immobilized FECH. Thus, in the three working channels of the biosensor, the monomeric form of FECH was immobilized. In the Fc4 channel, the FECH dimers were formed by injecting the FECH solution (5 μM) for 25 min (Appendix A).

Immediately after protein immobilization, the NHS/EDC mixture (10 μL 400 mM EDC and 10 μL 100 mM NHS) was injected for 2 min. It is necessary for chemical stabilization of the FECH dimers by forming additional covalent bonds between the two FECH subunits in the dimers (Figure 6A), between adjacent dimers (Figure 6B), and/or between the FECH subunits and the dextran matrix (Figure 6C). After chemical stabilization, the rate of spontaneous FECH dimers dissociation decreased from about 20 RU/min to <1 RU/min. To control the absence of a negative effect of this procedure on the ability of FECH to interact with other proteins, we used the Fc3 channel with FECH monomers, through which the NHS/EDC mixture was also injected. After that, FECH solutions were injected through channels Fc2 and Fc3 in the concentration range 0.2–3.6 μM (Figure 2). The set of obtained sensorgrams was used for calculation of K_d_ values of FECH dimerization. The treatment with NHS/EDC insignificantly influenced the K_d_ values (about 0.5 ± 0.1 μM). Thus, for further studies, all four biosensor channels were used: “empty” channel Fc1 (control 1) to correct the results for possible non-specific sorption of analytes on a dextran matrix; channel Fc2 with immobilized FECH monomers; channel Fc3 with immobilized FECH monomers treated with a mixture of NHS/EDC (control 2); channel Fc4 with immobilized chemically stabilized FECH dimers.

#### 4.2.3. The SPR Study of Protein–Protein Interactions (PPIs)

Solutions of FECH partner proteins (CYB5B, SMAD4, ADR) and FECH itself (concentration 5 μM) were injected (as analytes) through biosensor channels for 10 min at a flow rate of 10 μL/min. For this, the biosensor mode “1-2-3-4 Quickinject” was used. The control signal from the “empty channel” FC1 (without immobilized protein) was subtracted from signals, obtained from other flow channels with the immobilized FECH in monomeric and dimeric forms. Solution containing 2 M NaCl and 0.4% CHAPS was used for the regeneration of the chip surface by injection for 30 s at a flow rate of 50 μL/min. All the experiments were executed at least three times.

### 4.3. Computer Aided Molecular Modeling and Docking

The spatial structure of FECH (PDB ID 2qd1, chain B) and ADR (PDB ID 1cjc, chain A) were obtained from the RCSB Protein Data Bank (http://rcsb.org). The protein structures were optimized by Powell minimization using a Tripos force field in vacuum. The partial atomic charges on the proteins were calculated by Gasteiger–Huckel method. The structure of isatin was designed using Sybyl 8.1 software. The partial atomic charges of it was calculated by semiempirical AM1 methods using MOPAC 6.0. Protein–protein docking was performed using ClusPro [34], GRAMM-X [35] и ZDock [36] programs. Docking parameters were used as default. Forty variants of protein–protein complexes from these programs were selected for further analysis. Molecular docking of isatin to complexes FECH/ADR was performed using the DOCK 6.5 program [37]. The solvent-accessible surface of the target for docking was built based on the Connolly algorithm with a probe radius of 1.4 Å. The electrostatic and van der Waals potential fields generated over the target was calculated using a grid (spacing 0.3 Å); the non-bonded distance cutoff was 12 Å; the parameters for the van der Waals interactions were used from the dw_AMBER_parm99.defn set. The compounds were docked using a grid-based energy scoring option for minimization after their initial placement; the best docking pose was selected based on a scoring function from DOCK 6.5.

## Figures and Tables

**Figure 1 ijms-21-07605-f001:**
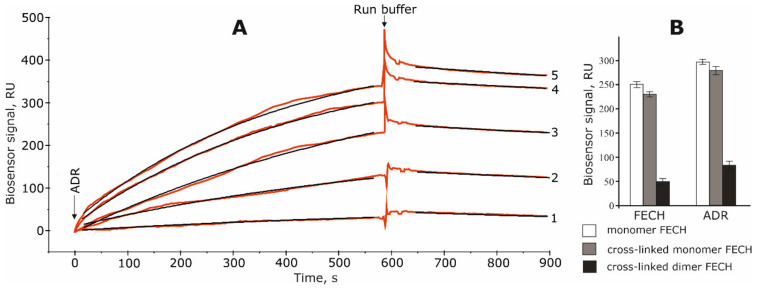
(**A**) Typical surface plasmon resonance (SPR) binding between immobilized ferrochelatase monomer on the optical chip and different concentration of adrenodoxine reductase (ADR): 0.1 μM (1), 0.5 μM (2), 1 μM (3), 2.5 μM (4), 5 μM (5). Fitting curves (theoretical models) are highlighted in black; Chi^2^ values are equal to 10.5 and 0.5 RU for association and dissociation phase, respectively. Complex dissociation constant (K_d_) values were calculated as ratio: K_d_ = k_off_/k_on_. (**B**) Binding levels of ferrochelatase (FECH) or ADR used as analytes with immobilized monomer (dimer) form of FECH protein. All analytes were injected at a concentration of 5 μM through biosensor channels for 10 min at a flow rate of 10 μL/min. Data represent mean ± SD, *n* = 3.

**Figure 2 ijms-21-07605-f002:**
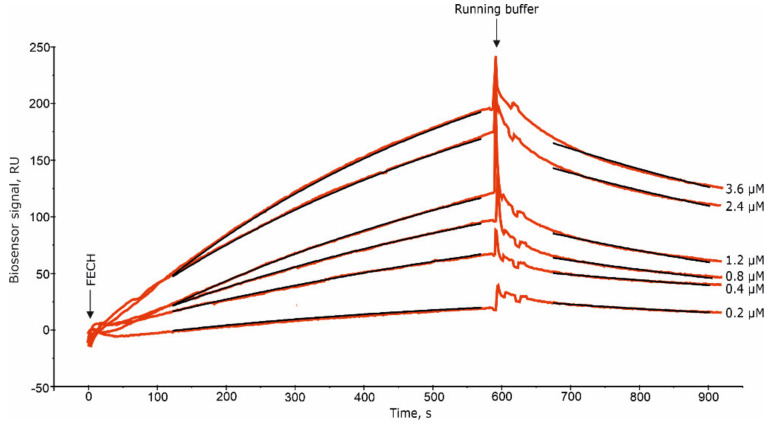
SPR analysis of ferrochelatase (FECH) dimerization. FECH solutions with different concentrations were injected through a biosensor channel with immobilized FECH monomers. Fitting curves (theoretical models) are highlighted in black; Chi^2^ values are equal to 0.3 and 1.0 RU for association and dissociation phase, respectively. Complex dissociation constant (K_d_) values was calculated as ratio: K_d_ = k_off_/k_on_.

**Figure 3 ijms-21-07605-f003:**
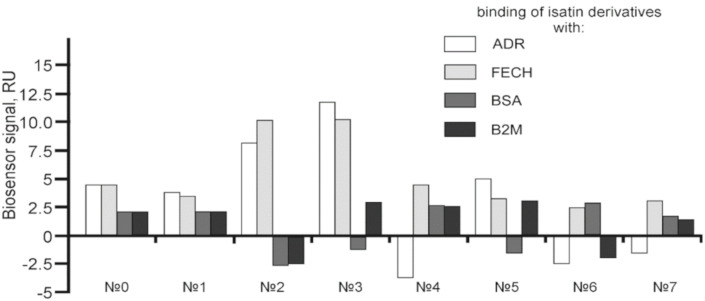
The average values of binding levels of isatin derivatives (100 µM) with immobilized on the CM5 optical chip FECH monomer form, ADR, B2M and BSA (*n* = 3). FECH, ADR and control proteins B2M and BSA were immobilized up to 4000 ± 200 RU levels using the standard amino-coupling protocol. Isatin and its derivatives were dissolved in 96% ethanol. Isatin derivatives samples in HBS-EP+ buffer containing 1% *v/v* ethanol were injected for 10 min at a flow rate of 10 µL/min. A solution containing 2M NaCl and 0.4% CHAPS was used for chip surface regeneration for 30 s at a flow rate of 25 µL/min. The biosensor signals of compound binding above 5 RU were considered to be significant (taking into consideration the baseline drift 0.5 RU/min after injection of the sample containing HBS-EP+ buffer with 1% ethanol). The numbers designate the following isatin derivatives tested: isatin (№0), 5-methylisatin (№1), 5-bromoisatin (№2), 5-Iodoisatin (№3), 5-fluoroisatin (№4), 5-nitroisatin (№5), 5,7-dichloroisatin (№6), 3-hydroxy-3-(2-oxopropyl)indolin-2-one (№7). Structures of these compounds are presented in Table 1.

**Figure 4 ijms-21-07605-f004:**
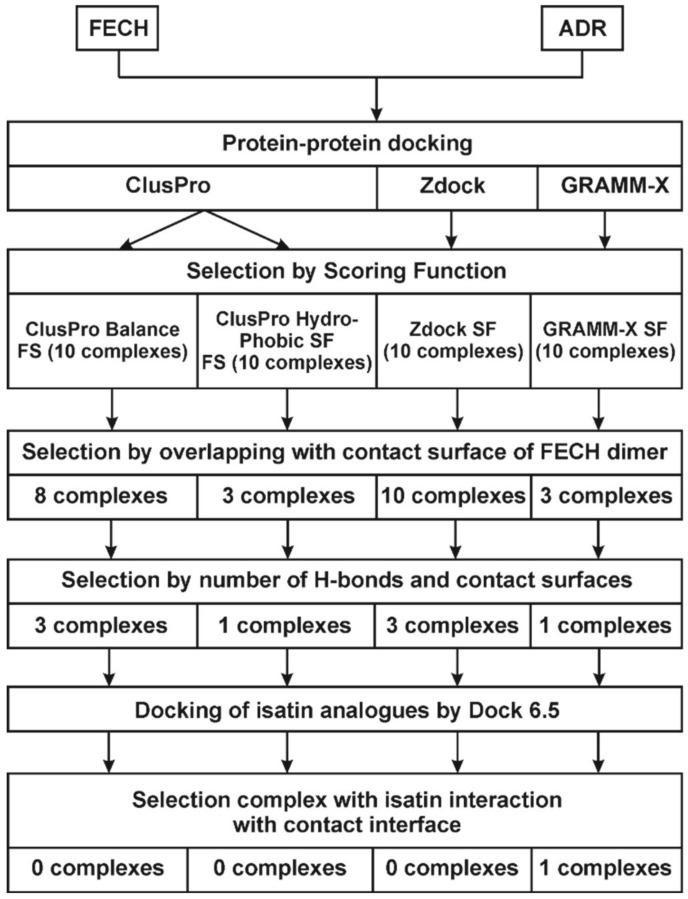
The algorithm of modeling of the FECH/ADR heterodimer complex with isatin.

**Figure 5 ijms-21-07605-f005:**
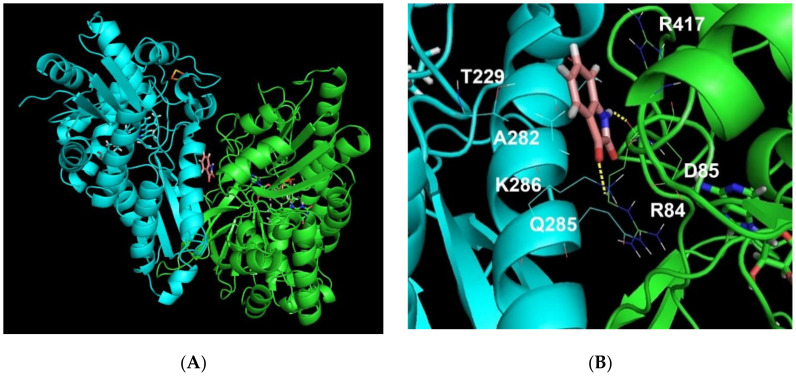
The model of a complex of FECH (cyan) and ADR (green) with isatin (**A**). Predicted isatin binding site in the FECH/ADR complex (**B**). Hydrogen bonds between isatin and amino acid residues of proteins are shown by a yellow dotted line. The model of complex was designed using FECH (PDB ID 2qd1, chain B) and ADR (PDB ID 1cjc, chain A).

**Figure 6 ijms-21-07605-f006:**
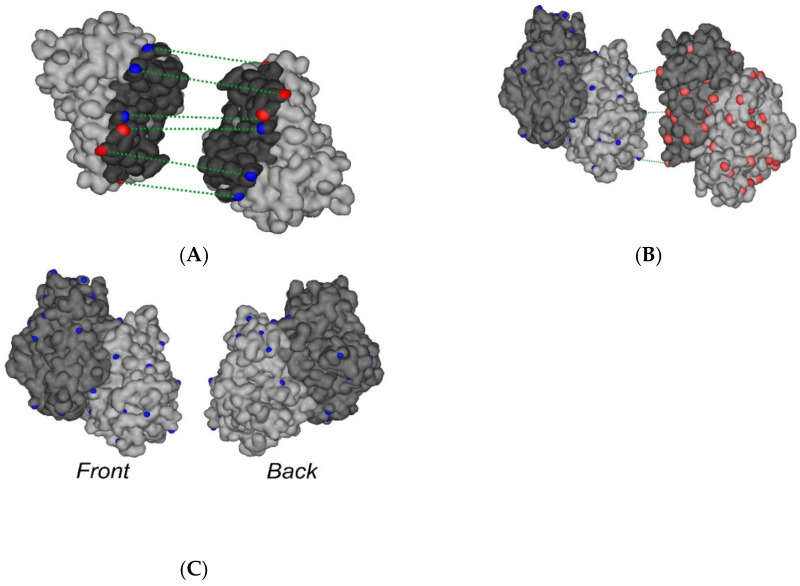
Visualization of 3D structure of ferrochelatase (FECH) protein (PDB ID 1HRK) with surface accessible chemical groups which can be critical for cross-linking between carboxyl groups of chip dextrane matrix and for intersubunit cross-linking. Surface lysine amino groups and carboxyl groups of ferrochelatase subunits are stressed with blue dots and red dots, respectively. Visualization was carried out using ViewerLite 5.0 software (Accelrys Inc., San Diego, CA, USA). (**A**) stabilization of FECH dimer via chemical cross-links between monomers in positions GLU255―LYS415, GLU292―LYS397, GLU289―LYS286, LYS286―GLU289, LYS397―GLU292 and LYS415―GLU255 is shown in dotted lines. The contact area of the monomers involved in the formation of a ferrochelatase homodimer is highlighted in dark gray. (**B**) two FECH dimer complexes with highlighted probable chemical cross-links “back to back” via amino groups of LYS (in positions 66, 106, 113, 133, 138, 145, 216, 243, 252, 320, 358, 379, 397) and carboxyl groups of GLU (in positions 80, 141, 149, 176, 251, 292, 359, 369, 413) and ASP (in positions 87, 246, 369, 383). In the current example, formation of three cross-links (LYS243―GLU149, LYS138―GLU369, LYS66—GLU251) is shown in dotted lines. The FECH monomer A is shown in light gray and the monomer B in dark gray color. (**C**) front and back view of the FECH dimer with marked lysine amino groups in positions 66, 106, 113, 133, 138, 145, 216, 243, 252, 320, 358, 379, 397 (for each subunits), which can form cross-linking with carboxyl groups of chip dextran matrix.

**Table 1 ijms-21-07605-t001:** The list of isatin derivatives which were used for the SPR analysis.

Compound Number	Name/CAS/Empirical Formula/Molecular Weight, Da	Structure
0	Isatin/91-56-5/C_8_H_5_NO_2_/147	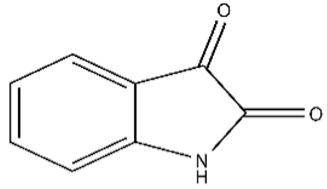
1	5-Methylisatin/608-05-9/C_9_H_7_NO_2_/161	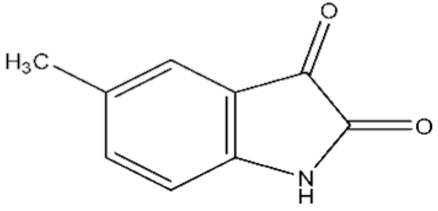
2	5-Bromoisatin/87-48-9/C_8_H_4_BrNO_2_/226	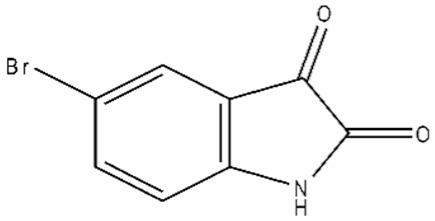
3	5-Iodoisatin/20780-76-1/C_8_H_4_INO_2_/273	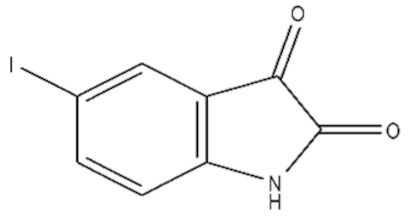
4	5-Fluoroisatin/443-69-6/C_8_H_4_FNO_2_/165	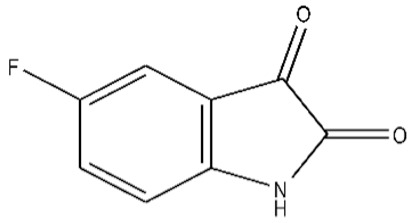
5	5-Nitroisatin/611-09-6/C_8_H_4_N_2_O_4_/192	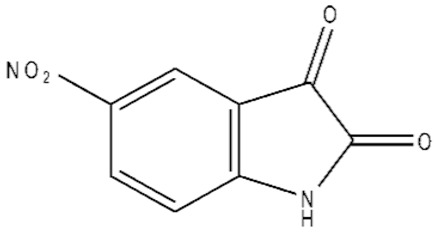
6	5,7-Dichloroisatin/6374-92-1/C_8_H_3_Cl_2_NO_2_/216	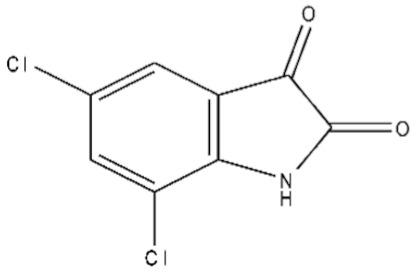
7	3-hydroxy-3-(2-oxopropyl)indolin-2-one/C_11_H_11_NO_3_/205	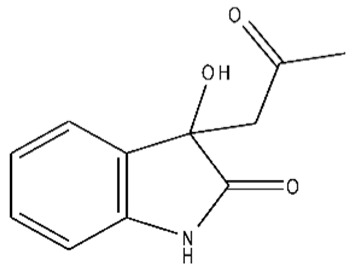

**Table 2 ijms-21-07605-t002:** K_d_ values (μM) of FECH/ADR and FECH/FECH complex formation in the presence of 100 μM isatin derivatives.

Control *	№0 **	№1 ***	№2	№3	№4	№5	№6	№7
**FECH/ADR**								
15.2 ± 1.2	3.4 ± 0.4	14.3 ± 1.2	15.5 ± 1.5	14.1 ± 0.8	13.8 ± 1.4	13.7 ± 0.9	14.7 ± 1.4	15.5 ± 0.8
**FECH/FECH**								
0.55 ± 0.11	0.60 ± 0.05	0.57 ± 0.06	0.53 ± 0.06	0.54 ± 0.05	0.61 ± 0.06	0.56 ± 0.08	0.58 ± 0.06	0.53 ± 0.10

* Control—without addition of a compound; ** №0—isatin; *** isatin derivatives are listed in Table 1.

**Table 3 ijms-21-07605-t003:** The values of the scoring function of ClusPro2.0 for models of the FECH/ADR heterodimer.

N of Complex	Balanced Scoring Function	N of Complex	Hydrophobic Scoring Function
1	−657.9	1	−931.3
2	−705.0	2	−910.8
3	−737.1	3	−808.0
4	−680.9	4	−986.8
5	−656.6	5	−815,3
6	−672.8	6	−922.9
7	−680.5	7	−830.9
8	−740.7	8	−914.8
9	−742,9	9	−915.6
10	−673.6	10	−882.2

**Table 4 ijms-21-07605-t004:** Preparation of the model system “monomer-dimer” FECH on the chip of a four-channel SPR biosensor.

Biosensor Channel	FECH Immobilization	Solution Injection		
		5 мM NaOH	FECH	NHS + EDC
Fc1 (control 1)	-	-	-	-
Fc2 (monomer form)	+	+	-	-
Fc3 (control 2)	+	+	-	+
Fc4 (dimer form)	+	+	+	+

Notes. Flow cell 1 (Fc1) is a control cell without any protein immobilization (control for nonspecific binding of analytes with dextran matrix); Fc2 is a cell with immobilized monomer form of FECH; Fc3 is a cell with immobilized monomer form of FECH stabilized by chemical cross-linking (additional control); Fc4 is a cell with rebuilt dimer form of FECH, stabilized by chemical cross-linking.

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
