# Peer review of "Mechanism of the Affinity-Enhancing Effect of Isatin on Human Ferrochelatase and Adrenodoxin Reductase Complex Formation: Implication for Protein Interactome Regulation"

_ijms, 2020, doi:10.3390/ijms21207605_

Round 1
Reviewer 1 Report
The authors have addressed all the comments.
Author Response
We are thankful for your constructive comments and suggestions which are valuable and helpful for revising and improving our manuscript.
Reviewer 2 Report
Although I appreciate the authors have attempted to answer my issues, I am not convinced.
- Although I am not aware of the Vasser lab software, I can say with surety that zdock DOES have scoring functions that are in the output files
- The SPR data information is very interesting. However, upon further investigation into these references, I note that many of the authors of these papers confirm my view that you cannot use this data alone. For example in Piepenbrink et al they state "The primary problem in low affinity titration curves lies in knowing where saturation is. Very simply, if the KD is the free ligand concentration where 50% binding occurs, how can one determine the concentration that gives 50% bound if the concentration that gives 100% bound cannot be determined or reliably estimated?...One method available in SPR is to independently determine the activity of the sensor surface and fix this value in subsequent analyses." The authors do not seem to have done these essential control experiments and I remain unconvinced that the SPR data alone is conclusive evidence of interactions.
Author Response
- Although I am not aware of the Vasser lab software, I can say with surety that zdock DOES have scoring functions that are in the output files
Our response:
The zdock scoring function has been included into a new Table S1 and S2 (see Supplementary Materials) and the following additions have been included in the text (see the Section 2.3):
Figure 4 shows the algorithm of modelling. ClusPro, GRAMM-X, and ZDock. Among these programs ClusPro and ZDock presented the values of scoring functions, while GRAMM-X presents top of complexes. Based on the scoring function of the ZDock output file (see Supplementary Materials, Tables S1, S2) 10 top ZDock possible complexes were selected. Since the ClusPro program calculates several scoring functions, two sets of 10 models with the best “balanced” and “hydrophobic” scoring functions were selected (Table 3). Thus, 40 models of possible complexes (10 from GRAMM-X, 10 from ZDock, and 20 from ClusPro) were selected for subsequent computer-aided analysis.
- The SPR data information is very interesting. However, upon further investigation into these references, I note that many of the authors of these papers confirm my view that you cannot use this data alone. For example in Piepenbrink et al they state "The primary problem in low affinity titration curves lies in knowing where saturation is. Very simply, if the KDis the free ligand concentration where 50% binding occurs, how can one determine the concentration that gives 50% bound if the concentration that gives 100% bound cannot be determined or reliably estimated?...One method available in SPR is to independently determine the activity of the sensor surface and fix this value in subsequent analyses." The authors do not seem to have done these essential control experiments and I remain unconvinced that the SPR data alone is conclusive evidence of interactions.
Our response:
The Piepenbrink’s approach is not applicable for our experiment protocol because “high-affinity” FECH and ADR do not exist.
The IJMS Academic Editor decision about our SPR data is ”SPR data do show an interaction”. According to commentaries of Academic Editor we have modified Figures 1 and 2; these figures now show simulated curves based on the Langmuir theoretical model of the 1:1 interaction performed in the separate calculation mode of BIAevaluation software. In addition, we modified description of these figures and added values of Chi2 parameter describing the fitting quality.
Figure 1. (A) Typical SPR binding between immobilized ferrochelatase monomer on the optical chip and different concentration of ADR: 0.1 μM (1), 0.5 μM (2), 1 μM (3), 2.5 μM (4), 5 μM (5). Fitting curves (theoretical models) are highlighted in black; Chi2 values are equal to 10.5 and 0.5 RU for association and dissociation phase, respectively. Complex dissociation constant (Kd) values was calculated as ratio: Kd = koff/kon. (B) Binding levels of FECH or ADR used as analytes with immobilized monomer (dimer) form of FECH protein. All analytes were injected at a concentration of 5 μM through biosensor channels for 10 min at a flow rate of 10 μL/min. Data represent mean ± SD, n = 3.
Figure 2. SPR analysis of ferrochelatase (FECH) dimerization. FECH solutions with different concentrations were injected through a biosensor channel with immobilized FECH monomers. Fitting curves (theoretical models) are highlighted in black; Chi2 values are equal to 0.3 and 1.0 RU for association and dissociation phase, respectively. Complex dissociation constant (Kd) values was calculated as ratio: Kd = koff/kon.
In addition we would like to point to the clear concentration-dependent behavior of sensorgrams, which definitely indicate existence of interaction between proteins.
This manuscript is a resubmission of an earlier submission. The following is a list of the peer review reports and author responses from that submission.
Round 1
Reviewer 1 Report
Ershov et al present the results of an SPR and molecular modelling/docking studies of the PPI between FECH and ADR which is stabilised by isatin. This work builds on the group’s previous paper (Ref 8) which also used SPR to determine the effect of isatin on the dissociation constants (Kd) of the FECH/ADR interaction. The key findings of the new SPR investigation outlined here is that isatin specifically induces this effect on FECH monomers that are able to form a binary complex with ADR. Thus, a ternary protein-ligand-protein complex is formed. The same effect is not observed for FECH dimers which do not interact with ADR. An in silico study presents a theoretical model for how isatin induces formation of this ternary complex.
Whilst these findings, and the theoretical binding model, are very interesting, I am of the opinion that it is not yet ready for publication. The following queries should be addressed and additional experiments performed where appropriate.
- The introduction should include a much clearer summary of the work in Ref 8 that compares the relevant Kds for the PPIs and the effect isatin has. This will bring greater clarity and focus to the new findings.
- Figure 1 – include concentrations used. I suggest that the data and discussion around SMAD4 and CYB5B are moved to an appendix.
- Effect of isatin and derivatives on FECH/ADR complexation (page 6, lines 5-12). The data alluded to in this paragraph is critical and its omission is puzzling. The manuscript must include a figure that shows data to support isatin having a specific stabilising effect FECH/ADR complexation, but not on FECH/FECH dimerization or ADR/FECHdimer binding. Likewise, it is important to show the isatin derivatives have no effect in comparison. This would in essence be an elaboration of the experiment conducted in Figure 1.
- Binding of isatin and derivatives to FECH and ADR etc. This data provides a good indication that isatin only interacts with FECH and ADR when the two are in complex, and thus formation of a ternary complex in a cooperative fashion. I suggest this data is presented after the complexation data (point 3). Again, I suggest focussing on FECH and ADR, moving the other data to an appendix. It would be very convincing if the cross-linking approach could be used here to fuse the FECH/ADR complex, either as an immobilised construct or in solution, and then run the isatin/aminoisatin binding experiments. Assuming cross-linking does not perturb the interface, one would expect to see a notable increase in RU to support this hypothesis. Perhaps even an opportunity to extract a Kd. This additional experimental data is not a requirement for publication, but in my opinion its inclusion would enhance the scientific quality of the work significantly.
- 5-bromo and 5-iodo isotin derivatives show strong binding to FECH and ADR in isolation but apparently show no effect on the FECH/ADR PPI. The specificity of isatin for the interface pocket compared to these derivatives should be rationalised using the molecular docking experiments. Can the in silico tools available be used to propose an alternative binding site if an alternative theory cannot be proposed otherwise?
- A supplementary information document providing all relevant SPR sensorgrams should be provided.
- A thorough check for quality of English and grammatical errors is needed.
Author Response
Point 1:
The introduction should include a much clearer summary of the work in Ref 8 that compares the relevant Kds for the PPIs and the effect isatin has. This will bring greater clarity and focus to the new findings.
Response 1:
The following information has been added to the Introduction section:
“Our previous SPR study showed that a pharmacologically relevant concertation of isatin (100 µM) significantly increased affinity of the FECH/ADR complex: the Kd value decreased from 13 µM to 2 µM [8]. These isatin-induced changes in ∆Kd included both an increase in the FECH/ADR association rate (the kon value demonstrated a four-fold increase) and a less than 2-fold decrease in the dissociation rate constant (koff) [8]. At a higher pharmacological concentration of isatin (270 μM) this effect became even more pronounced (the change in the Kd value was more than 15 times).”
Point 2:
Figure 1 – include concentrations used.
Response 2:
The following information has been added:
“all analytes (CYB5B, SMAD4, ADR and FECH) were injected at a concentration of 5 μM through biosensor channels for 10 min at a flow rate of 10 μL/min”
Point 3:
I suggest that the data and discussion around SMAD4 and CYB5B are moved to an appendix.
Response 3:
Although our paper is focused on the FECH/ADR complex formation and the effect of isatin on this process, data about other two other proteins (SMAD4 and CYB5B) demonstrate selectivity of ADR interaction only with FECH monomers. Other proteins do not demonstrate such selectivity and bind to FECH monomers and dimers.
Point 4:
Effect of isatin and derivatives on FECH/ADR complexation (page 6, lines 5-12). The data alluded to in this paragraph is critical and its omission is puzzling. The manuscript must include a figure that shows data to support isatin having a specific stabilising effect FECH/ADR complexation, but not on FECH/FECH dimerization or ADR/FECHdimer binding. Likewise, it is important to show the isatin derivatives have no effect in comparison. This would in essence be an elaboration of the experiment conducted in Figure 1.
Response 4:
This study is a logic continuation of our previous work where details of FECH/ADR complex formation in the presence and the absence of isatin are given (Ershov et al 2017, Ref 8; https://doi.org/10.1002/pro.3300]. In this study we have demonstrated that isatin and its analogues did not influence kinetic parameters of FECH dimerization (Fig 4A) and FECH/ADR heterodimerization. Isatin derivatives (100 µM) were tested in the biosensor system of FECH/ADR и FECH/FECH complex formation described in [Ershov et al 2017]). This information was added to the Appendix as Table A1 and Figure S1 (Supplementary file). In addition, isatin did not influence the ADR/FECH dimer complex (Figure 1). We did not find any significant effect of isatin derivatives on kinetics and affinity of FECH/FECH and FECH/ADR complex formation as compared with control (without isatin derivatives). This suggests specificity of the isatin effect on FECH/ADR complex formation.
In addition to Table A1, the following information has been included to the bottom paragraph of the section 2.2 (page 6):
[8]. We did not find significant differences in the kinetic and affinity parameters of FECH/ADR and FECH/FECH complex formation in the presence of seven different isatin derivatives as compared to control (without isatin derivatives) (Table A1), whereas isatin induced complexation of FECH/ADR (Figure S1 and also [8]), but did not affect the formation of FECH/FECH dimers. Although parameters of complex formation between FECH (cross-linked dimer) and ADR (Figure 1) were not determined, but SPR protein binding signals in the presence of isatin remained unchanged.
Point 5:
Binding of isatin and derivatives to FECH and ADR etc. This data provides a good indication that isatin only interacts with FECH and ADR when the two are in complex, and thus formation of a ternary complex in a cooperative fashion. I suggest this data is presented after the complexation data (point 3). Again, I suggest focussing on FECH and ADR, moving the other data to an appendix. It would be very convincing if the cross-linking approach could be used here to fuse the FECH/ADR complex, either as an immobilised construct or in solution, and then run the isatin/aminoisatin binding experiments. Assuming cross-linking does not perturb the interface, one would expect to see a notable increase in RU to support this hypothesis. Perhaps even an opportunity to extract a Kd. This additional experimental data is not a requirement for publication, but in my opinion its inclusion would enhance the scientific quality of the work significantly.
Response 5:
We thank for the valuable suggestion, which we plan to investigate experimentally in the recent future. This study has been undertaken for the Special Issue “Biological Activities and Biomedical Application of Isatin and its Analogues “. Its deadline is August 31, 2020.
Point 6:
5-bromo and 5-iodo isotin derivatives show strong binding to FECH and ADR in isolation but apparently show no effect on the FECH/ADR PPI. The specificity of isatin for the interface pocket compared to these derivatives should be rationalized using the molecular docking experiments. Can the in silico tools available be used to propose an alternative binding site if an alternative theory cannot be proposed otherwise?
Response 6:
Indeed, despite binding of 5-bromo and 5-iodo isatins to proteins, molecular docking did not predict models of 5-bromo isatin and 5-iodo isatin binding to the isatin binding pocket (the interface area of the FECH/ADR complex).
This study was focused on isatin as a functionally important endogenous/pharmacological regulator and isatin derivatives were used as controls for specificity of the isatin effects on the affinity of complex formation and as negative controls for molecular docking. Isatin derivatives are traditionally considered as potential drug-like agents acting on many protein targets. The investigation of this aspect as well as the mapping of potential binding sites for 5-bromo isatin and 5-iodo isatin in 3D protein models is out of goals of this study and it needs specials consideration.
In this context it should be noted that all docking programs operating on different algorithms give out a huge number of structural hypotheses for protein-ligand docking. The choice of the most probable variants from this set of hypotheses is basically impossible without a combination with the results of special experimental studies.
Point 7:
A supplementary information document providing all relevant SPR sensorgrams should be provided.
Response 7:
SPR sensorgrams has been added (See the Supplementary file).
Point 8:
A thorough check for quality of English and grammatical errors is needed.
Response 8:
We have corrected our English language and style.
Reviewer 2 Report
I enjoyed reading this manuscript which describes SPR analysis of FECH with specific proteins and the effect of isatin's on this binding. This was well supported by computational modelling which used the biological data to guide the models. I have no issues what-so-ever with this computational modelling and appreciate this is how modelling should be conducted. My only suggestion to strengthen this would be to look at programs such as PISA (https://www.ebi.ac.uk/pdbe/pisa/) which can give you insightful information regarding if you model is suggestive of a "real" PPI interface. It would be great to have this data and the model data i.e. "scores" as a table for those who are interested.
My more significant issues are with the SPR data. I am concerned that you cannot really distinguish between real binding an non-specific binding using a single concentration. All of these experiments really need to be conducted in a dose-response manner (5 concentrations minimum).
Also please change fig 2 as I'm assuming you mean BA, not albumin?
Author Response
Point 1:
I enjoyed reading this manuscript which describes SPR analysis of FECH with specific proteins and the effect of isatin's on this binding. This was well supported by computational modelling which used the biological data to guide the models. I have no issues what-so-ever with this computational modelling and appreciate this is how modelling should be conducted. My only suggestion to strengthen this would be to look at programs such as PISA (https://www.ebi.ac.uk/pdbe/pisa/) which can give you insightful information regarding if you model is suggestive of a "real" PPI interface. It would be great to have this data and the model data i.e. "scores" as a table for those who are interested.
Response 1:
We have made pilot prediction of the designed complex using the PISA program. According to the scoring function of this program this prediction does not look as a native protein-protein complex. But there are several challenges for correctness of this output. First, the ternary complex of FECH/ADR/isatin is a temporary formed complex without known physiological function. It should be noted that most protein-protein complexes in PDB (source of training set for such programs) are differed from our case. At present without special experiments it is difficult to estimate the correctness of the PISA scoring function for complexes like FECH/ADR/isatin. The second question is application of such scoring function to complexes designed using rigid protein-protein docking. This also needs additional experiments including structure optimization by molecular dynamics etc.
Point 2:
My more significant issues are with the SPR data. I am concerned that you cannot really distinguish between real binding an non-specific binding using a single concentration. All of these experiments really need to be conducted in a dose-response manner (5 concentrations minimum).
Response 2:
The concentration-dependent mode of the interaction between ADR, CYB5, SMAD4 proteins with ferrochelatase was investigated in our previous study (Ref. 8, https://doi.org/10.1002/pro.3300): using the SPR technology we found the dose-response behavior for interaction of the studied proteins and calculated parameters of this interaction.
For each novel interaction we use at least 5 concentrations of corresponding analytes. For example, in experiments on FECH homodimerization we used 6 concentrations of the analyte (see Figure A4).
Point 3:
Also please change fig 2 as I'm assuming you mean BA, not albumin?
Response 3:
Figure 2 has been corrected.
Round 2
Reviewer 1 Report
The authors have addressed all the comments.
Reviewer 2 Report
Unfortunately the authors did not take on any of my suggestions.
Modelling - I appreciate the intricacies with PISA and the authors would not have had to use those scores. However according to the methods, the authors used 3 alternate docking programs. Even though scoring functions do have issues, these scores should have been easy to put into a table for those who are interested in these sorts of things.
SPR - the authors added no extra data. The only dose-response binding data is Appendix A1 (which has only 4 concentrations and has not reached equilibrium) and Appendix A4 (which again does not look like clear dose-response binding and the slopes of those curves are not high quality). This does not cover all the new binding experiments they have added to this manuscript. I would also suggest the authors take a look at the following guide for SPR which will enable them to understand what a quality SPR curve should look like https://www.sprpages.nl/sensorgram-tutorial/a-sensorgram